# A Novel Catalytically Inactive Construct of Botulinum Neurotoxin A (BoNT/A) Directly Inhibits Visceral Sensory Signalling

**DOI:** 10.3390/toxins16010030

**Published:** 2024-01-07

**Authors:** Hodan Ibrahim, Kevin Retailleau, Fraser Hornby, Jacquie Maignel, Matthew Beard, Donna Marie Daly

**Affiliations:** 1School of Pharmacy and Biomedical Sciences, University of Central Lancashire, Preston Campus, Preston PR1 2HE, UK; 2Ipsen, Abingdon OX14 4RY, UK; fraser.hornby@ipsen.com (F.H.); matthew.beard@ipsen.com (M.B.); 3Ipsen, 5 av Canada, 91940 Les Ulis, France; kevin.retailleau@ipsen.com (K.R.); jacquie.maignel@ipsen.com (J.M.)

**Keywords:** botulinum toxin, sensory, bladder, nerve, mechanosensitivity

## Abstract

Botulinum neurotoxin A (BoNT/A) is a potent neurotoxin that silences cholinergic neurotransmission through the cleavage of the synaptic protein SNAP-25. Previous studies have shown that, in addition to its paralytic effects, BoNT/A can inhibit sensory nerve activity. The aim of this study was to identify how BoNT/A inhibits afferent signalling from the bladder. To investigate the role of SNAP-25 cleavage in the previously reported BoNT/A-dependent inhibition of sensory signalling, we developed a recombinant form of BoNT/A with an inactive light chain, rBoNT/A (0), unable to paralyse muscle. We also developed recombinant light chain (LC)-domain-only proteins to better understand the entry mechanisms, as the heavy chain (HC) of the protein is responsible for the internalisation of the light chain. We found that, despite a lack of catalytic activity, rBoNT/A (0) potently inhibited the afferent responses to bladder distension to a greater degree than catalytically active rBoNT/A. This was also clear from the testing of the LC-only proteins, as the inactive rLC/A (0) protein inhibited afferent responses significantly more than the active rLC/A protein. Immunohistochemistry for cleaved SNAP-25 was negative, and purinergic and nitrergic antagonists partially and totally reversed the sensory inhibition, respectively. These data suggest that the BoNT/A inhibition of sensory nerve activity in this assay is not due to the classical well-characterised ‘double-receptor’ mechanism of BoNT/A, is independent of SNAP25 cleavage and involves nitrergic and purinergic signalling mechanisms.

## 1. Introduction

Botulinum neurotoxin serotype A (BoNT/A) is a member of a family of potent neurotoxins produced by the bacteria Clostridium botulinum (see [1] for review). These neurotoxins are the cause of botulism, a potentially lethal disease resulting in muscle paralysis and asphyxiation [2]. The BoNT/A protein is composed of a heavy chain (HC) of 100 kDa and a light chain (LC) of 50 kDa, each with functionally distinct properties. The HC contains two functional domains, a receptor-targeting domain and a translocation domain. The receptor-targeting domain preferentially targets cholinergic neurons, and, once bound, the translocation domain translocates the LC into the cytoplasm. The paralytic effect of BoNT/A has been well documented and is primarily due to the ability of the LC to bind to and cleave the intracellular SNAP-25 protein, a crucial member of the soluble N-ethylmaleimide-sensitive fusion protein (NSF) attachment protein receptor (SNARE) complex that mediates neurotransmitter release. This leads to the inhibition of the release of acetylcholine (ACh) at the neuromuscular junction, resulting in sustained muscle paralysis. Despite its prominence as a potent toxin, BoNT/A has been used widely in the clinic for cosmetic and medical indications, where its paralytic effects are exploited to provide targeted inhibition of muscle contraction [3,4].

In recent years, BoNT/A formulations have been approved as a treatment for overactive bladder syndrome (OABS), a condition characterised by the symptoms of urinary urgency with or without incontinence and frequent urination. BoNT/A was initially believed to provide relief for OABS by inhibiting the contractility of the bladder smooth muscle via an effect at the presynaptic/parasympathetic level. However, the diagnostic criteria for OABS are largely dependent on the presence of urinary urgency [5], a symptom that is believed to be caused by the hyperactivity of peripheral afferent pathways or by central sensitisation leading to a heightened perception of peripheral input to the brain, rather than muscle contraction [6].

Clinical studies investigating the efficacy of BoNT/A in the treatment of OABS have shown significant reductions in the incidence of urgency [7,8,9,10,11], indicating that, in the lower urinary tract (and potentially elsewhere), BoNT/A may work via a non-classical mechanism, which might not involve the perturbation of ACh release. An experimental study using mouse tissue has shown that BoNT/A can directly inhibit the sensory responses of intramural afferent nerves from the mouse bladder [12]. Antinociceptive effects unrelated to muscle paralysis have also been reported in the literature, where BoNT/A treatment caused a significant reduction in formalin- or carrageenan-evoked pain responses of the rat [13,14].This has been further confirmed by experimental findings that show a significant inhibition of trigeminal neurons following BoNT/A treatment [15,16]. While the effect of BoNT/A on muscle contractility is well documented and understood, evidence from these studies points to a direct sensory effect of BoNT/A that is not sufficiently explained by the inhibition of ACh release at neuromuscular junctions.

The aim of this study was to understand the mechanisms involved in the BoNT/A-mediated modulation of sensory signalling. For the first time, we show that the significant inhibition of afferent nerve signalling was achieved without evidence of cleaved SNAP-25 and by molecules that did not include the receptor binding or translocation domains of BoNT/A. These findings propose a novel mechanism by which BoNT/A may inhibit visceral hypersensitivity without causing muscle paralysis.

## 2. Results

### 2.1. AboBoNT-A Directly Inhibited Sensory Nerve Firing from the Bladder

To investigate the effect of 100 U/mL aboBoNT-A on the ability of intramural afferent nerves to detect physiological and nociceptive levels of bladder distension, we used a well-characterised ex vivo preparation that facilitated the concomitant recording of intravesical pressure and afferent nerve firing. The distension of the bladder to 50 mmHg caused a graded increase in afferent nerve discharge consistent with the recruitment of a number of mechanosensitive pelvic and hypogastric afferent nerve units. This typical afferent response was significantly blunted following treatment with 100 U/mL aboBoNT-A. Figure 1A shows a reduction in the mechanosensitive responses to filling 90 min after treatment with 100 U/mL aboBoNT-A. AboBoNT-A significantly reduced mechanosensitive afferent firing over the 90 min protocol (Figure 1B). To explore whether there was a temporal element in the aboBoNT-A-induced inhibition, the levels of inhibition at the measured time points were compared. It was found that, 30 min after treatment, peak nerve firing at 50 mmHg was reduced by 18.2% (±6.23%), while by 90 min post-treatment, nerve firing was reduced by 32.5% (±10.7%). However, the data showed no difference between the action of aboBoNT-A on units firing at physiological pressure (15 mmHg) and at nociceptive pressure (50 mmHg) (Figure 1C). Bladder compliance appeared to be slightly increased following aboBoNT-A treatment (Figure 1D).

### 2.2. Investigation of the Presence of SNAP-25 and cSNAP-25 within the Bladder Wall

Using immunohistochemistry, the expression of SNAP-25 and the cleaved form of SNAP-25 (cSNAP-25) that remains following BoNT/A cleavage was investigated.

As shown in Figure 2, SNAP25 immunoreactivity (SNAP-25-IR) was found throughout the detrusor smooth muscle layers of the bladder wall in a staining pattern reminiscent of nerve fibres. The level of staining was similar between the vehicle-treated control preparations and the aboBoNT-A-treated preparations, which suggests that aboBoNT-A did not affect the expression of the full-length SNAP-25 protein. 

There was no detectable cSNAP25 immunoreactivity in either the aboBoNT-A-treated or vehicle-treated control bladder tissue (Figure 3). The cSNAP-25 antibody was validated as being able to detect cleaved SNAP25 in tissue sections, against sections of rat skeletal muscle from animals previously treated with the aboBoNT-A toxin (see Appendix A). 

### 2.3. Catalytically Inactive BoNT/A Significantly Inhibited Sensory Nerve Firing from the Bladder

To further investigate whether SNAP-25 cleavage was necessary for the BoNT/A effect on bladder sensory nerves, recombinant catalytically active (rBoNT/A) and catalytically (rBoNT/A(0)) forms of the toxin were generated; rBoNT/A(0) contains the point mutations E224Q and H227Y, which disrupt the coordination of the active site zinc ion and abolish measurable endopeptidase activity [17,18]. We first tested the recombinant toxins on mouse skeletal muscle contractility, using the well-established phrenic nerve hemidiaphragm assay. After incubation with the catalytically active rBoNT/A, the contraction amplitude of the diaphragm was completely inhibited, with a t50 of 24.1 (+/− 0.1) minutes consistent with a cholinergic blockade of the neuromuscular junction caused by SNAP25 cleavage, which has been well documented. In contrast, the application of the catalytically inactive construct rBoNT/A (0) had no effect on skeletal muscle contractility, as there was no difference in contraction amplitude to the buffer control, which was similar to preincubation contraction (Figure 4). 

Preincubation developed force was inhibited by 99.45% (+/−0.06%) by rBoNT/A1. After three hours of incubation, there was no difference between the effect of rBoNT/A (0) and the buffer control (n = 3) on contraction. 

The recombinant toxins were then tested in a bladder assay. The representative traces in Figure 5A,B show that, in the extracellular afferent nerve recordings, rBoNT/A produced a moderate inhibition of nerve firing, similar to that achieved by aboBoNT-A (Figure 1A). This was reflected in the analysis, which showed that 71.5% (+/−9.8%) of afferent firing remained 90 min after treatment with rBoNT/A; this was significant compared to the vehicle-treated preparations (Figure 5C). Remarkably, however, the catalytically inactive rBoNT/A (0) construct induced a profound reduction in the mechanosensitive afferent responses to distension, with 26.02% (+/−5.72%) of the afferent response remaining 90 min after treatment (Figure 5B,C). Rather than showing an attenuated effect (as might be reasonably expected due to the inability of rBoNT/A (0) to cleave SNAP-25), rBoNT/A (0) inhibited sensory responses more potently than its catalytically active counterpart rBoNT/A (Figure 5C). 

While the rBoNT/A-treated preparations did not affect the compliance of the bladder compared to the vehicle-treated preparations, rBoNT/A (0) caused a small but significant reduction in bladder compliance (Figure 5D).

### 2.4. The Light Chains of rBoNT/A and rBoNT/A(0) Alone Inhibited Sensory Signalling from the Bladder

Following the finding that mechanosensory signalling was significantly reduced without the cleavage of SNAP-25, we further explored the role of the structure of the BoNT/A molecule by testing the light chain alone. The catalytically active (Figure 6A) and inactive (Figure 6B) LC proteins inhibited distension-induced firing 90 min after treatment. Like the full-length proteins characterised in Figure 6, the rLC/A (0) protein inhibited afferent firing significantly more potently than the rLC/A protein (Figure 6C). There appeared to be a differential effect on the pressure–volume relationship, as rLC/A (0) was not different to the vehicle-treated preparations, while rLC/A significantly reduced the pressure–volume relationship (Figure 6D). These findings suggest that the receptor binding and translocation domains of BoNT/A (HC/A) are not necessary for the sensory inhibition produced by BoNT/A, and they suggest that the sensory effect is dependent solely on the light chain of the toxin.

To confirm that the observed effect on bladder mechanosensation was due to the activity induced by the light chain domain of BoNT/A, we next investigated the effect of the heavy chain only (HC/A). The application of 3.6pM HC/A did not alter bladder mechanosensitivity (as shown in Figure 7), as after 90 min post-treatment, 89.03% (+/−3.032%) of distension-induced nerve firing remained (*p* = 0.9819; n = 4). The pressure–volume relationship also remained unchanged 90 min after treatment with HC/A (*p* = 0.0513; n = 4). Peak firing at 15 and 50 mmHg did not appear to be altered over time (*p* = 0.0927; n = 4).

### 2.5. The Inhibitory Effect of rBoNT/A (0) Was Reversed by Nitrergic and Purinergic Antagonists

We applied the non-selective nitric oxide synthase (NOS) inhibitor L-NAME at 1mM prior to and alongside rBoNT/A (0) to assess its effect on bladder mechanosensation (Figure 8A). L-NAME alone did not alter distension-induced firing; however, when applied alongside BoNT/A (0), L-NAME reversed the potent BoNT/A (0) inhibition and led to a slight (ns) increase in mechanosensory nerve firing (Figure 8B). With regard to the pressure–volume relationship, L-NAME alone had no effect (Figure 8C), while co-application with BoNT/A (0) revealed a significant increase in bladder compliance (Figure 8D).

The P2X receptor antagonist 2′,3′-O-(2,4,6-Trinitrophenyl) adenosine-5′-triphosphate tetra (triethylammonium) salt (TNP-ATP) at 30 μM did not affect distension-induced afferent signalling (Figure 9A) or the pressure–volume relationship (Figure 9C). When TNP-ATP was applied alongside rBoNT/A(0), the potent BoNT/A(0) inhibition was partially dampened. Afferent firing was significantly lower than that of the control but not reduced to the same degree as when reduced by BoNT/A (0) alone (Figure 9B). The pressure–volume relationship was significantly increased by the presence of TNP-ATP alongside BoNT/A (0) (Figure 9D).

## 3. Discussion

In the present study, we show that a catalytically inactive BoNT/A construct caused a profound inhibition of sensory nerve firing from the bladder. Strikingly, we show that the magnitude of the inhibitory effect was greater than that produced by a catalytically active recombinant BoNT/A toxin (rBoNT/A) or the full BoNT/A complex (i.e., aboBoNT-A). This unexpected discovery may have important implications regarding the clinical efficacy and potency of BoNT/A, particularly in conditions where sensory inhibition is beneficial (i.e., pain and/or hypersensitivity). Moreover, we also show that both the recombinant light chain (rLC/A) and the catalytically inactive recombinant light chain (rLC/A (0)) exhibited the capability to significantly reduce afferent signalling, despite the absence of the typical heavy chain responsible for cell binding and entry. This suggests that the inhibition of sensory nerves by BoNT/A may not be solely dependent on SNAP25 cleavage but may involve other activities within the light chain region of the molecule.

### 3.1. Abobotulinumtoxin A Directly Inhibited Sensory Nerve Signalling

Abobotulinumtoxin A (aboBoNT-A) is a clinically available formulation of the BoNT/A complex that has been shown to be effective in reducing the incidence of urgency and frequency in clinical studies of OAB patients [9,19];. However, it is not clear whether aboBoNT-A can act directly on afferent nerves innervating the bladder to inhibit mechanosensation, an important component of micturition regulation, the disruption of which is thought to be involved in the development of urgency. In this study, we show that the intravesical instillation of aboBoNT-A into the mouse bladder resulted in a notable decrease in distension-induced afferent firing, consistent with previous findings reported [12]. However, the precise mechanisms underlying the BoNT/A-mediated modulation of bladder sensory signalling remain elusive [12]. 

As SNAP-25 is the best described target of BoNT/A, we performed immunohistochemistry to look at SNAP-25 immunoreactivity in aboBoNT-A-treated bladders. Our experiments revealed abundant SNAP-25 immunoreactivity throughout the sub-epithelial and intramuscular nerve fibres of the bladder. This is in agreement with previous studies and confirms the presence of SNAP-25 in the mouse bladder [20,21,22];. Interestingly, however, we found no immunoreactivity for cleaved SNAP-25 (cSNAP-25) in bladders that received the intravesical instillation of aboBoNT-A, despite seeing a clear inhibition of nerve firing; this suggests that either SNAP-25 was not cleaved in these experiments or any cleavage was below the level of detection of this method. The method of aboBoNT-A application might go some way in providing an explanation for this. When aboBoNT-A is used clinically, it is typically injected into the treatment site; however, in this study, we intravesically infused the toxin into the bladder. In a comparative study of BoNT/A application methods, Chuang, (2009), also found that instillation within the bladder did not lead to the immunostaining of cSNAP-25, whereas an injection into the bladder wall did [21]. The lack of cSNAP-25 immunoreactivity within deeper structures within the bladder wall such as the area under the epithelium (known as the suburothelium) and the nerve fibres that lie within the muscle layers following instillation might suggest that aboBoNT-A was restricted to the superficial areas of the bladder wall (i.e., the apical epithelial cells) or that it was working via a non-canonical pathway. It is important to emphasise that these data do not exclude the possibility that SNAP-25 cleavage did occur but that the level was below the limits of detection for our antibody. Further studies are required to irrefutably confirm that aboBoNT-A did not cleave SNAP-25 in this assay using other more sensitive methodologies. 

### 3.2. The Effect of Catalytically Inactive BoNT/A on Bladder Mechanosensitivity

The catalytic activity of BoNTs is dependent on the Zn^2+^ endopeptidase HEXHH motifs within the light chain [23], which have been targeted in previous studies to produce BoNT constructs with little to no catalytic activity [23,24,25]. Enzymatic activity within the light chain was disrupted through the substitution of essential residues within the HEXHH motif that interact with the Zn^2+^ ion [26]. In the present study, the E224Q/H227Y mutations were made to render the rBoNT/A (0) and rLC/A (0) proteins catalytically inactive [17], which was confirmed using a phrenic nerve hemidiaphragm assay (Figure 4). Contrary to what might be expected, rBoNT/A (0) was significantly more effective at inhibiting afferent firing than active rBoNT/A. This suggests that, when BoNT/A has been rendered catalytically inactive, not only is it able to inhibit sensory nerves, but also the inhibition produced is greater than that for the catalytically active protein. This is particularly surprising given that the only difference between the two compounds was the aforementioned mutation (E224Q/H227Y). 

It is possible that rBoNT/A (0) may still be able to enter nerves and disrupt the activity of SNAP-25 despite its inability to cleave it, resulting in a reduced exocytotic release of sensory mediators. However, this does not explain the gulf between the efficacy of the two compounds, especially as the cleavage induced by the active neurotoxin would presumably arrest the function of SNAP-25 more effectively. Previous studies have shown SNAP-25 activity to be unaffected by inactive forms of BoNT/A. Baskaran et al. (2013) found that, while the deactivated recombinant BoNT/A (drBoNT/A) was able to enter motor nerve terminals, it did not affect the release of acetylcholine nor did it inhibit the mouse toe spread reflex [24]. Ravichandran et al. (2015) conducted an in vivo LD50 toxicity assay to characterise their inactive BoNT/A construct (E224A/E262A), finding that it was unable to cleave SNAP-25, as well as being 1.2 million times less toxic than wild-type BoNT/A [27]. These findings suggest that the function of SNAP-25 is unaffected by an inactive form of BoNT/A and that any effects are more likely due to SNAP-25-independent activity. 

Although SNAP-25 is BoNT/A’s preferential target, it has been shown to cleave murine SNAP-23 at high concentrations in a proteomic cleavage assay and in rat kidney cells [28,29]. In the bladder, Hanna-Mitchell et al. (2015) reported a decrease in SNAP-23 staining post-BoNT/A treatment in rat uroepithelial cells, suggesting the cleavage of SNAP-23 [30]. Thus, one possibility is that, on the apical surface of the bladder, aboBoNT-A targets SNAP-23 rather than SNAP-25, resulting in sensory inhibition. However, the mutation in rBoNT/A (0), which prevents the molecule from cleaving SNAP-25, disrupts the coordination of the catalytic Zn^2+^ ion and removes any proteolytic cleavage ability; thus, it is reasonable to assume that rBoNT/A (0) would therefore also not be able to cleave SNAP-23.

The potent inhibition induced by rBoNT/A (0) was reversed by co-application with purinergic and nitrergic antagonists.

As the rBoNT/A and rBoNT/A (0) compounds are near-identical apart from the substitution of two amino acids, it is reasonable to assume that any effects aside from SNAP-25 cleavage and proteolytic activity may remain in rBoNT/A (0). In the bladder, sensory mediators such as adenosine triphosphate (ATP) and nitric oxide (NO) have been shown to play a role in the modulation of the afferent responses to stretch [31,32,33,34]. In the presence of BoNT/A, the release of these mediators has been shown to be altered [12,30,35]. While the release of ATP in the bladder has been shown to occur partly via exocytosis through vesicles and partly through pannexin/connexin hemichannels and other means [36,37,38], NO release is wholly non-vesicular [34]. 

By investigating these sensory pathways using purinergic and nitrergic antagonists, we aimed to better understand the SNAP-25-independent actions of rBoNT/A (0). While L-NAME alone did not have any effect on distension-induced afferent firing, when co-applied with rBoNT/A (0), the potent inhibitory effect of BoNT/A(0) was reversed to the point where a slight hypersensitivity to distension was seen. Increased NO release has been shown to be inhibitory to afferent firing [39], while in the present study, L-NAME alone did not affect distension responses. This finding is supported by Yu & De Groat (2013), who found that L-NAME alone did not alter afferent firing in basal conditions but reversed L-arginine-dependent inhibition [40]. It may be possible that rBoNT/A (0) caused NO to be released at a level great enough to potently inhibit afferent firing, which would explain the reversal following L-NAME co-application. 

To investigate the purinergic pathway, we tested the P2X antagonist TNP-ATP and its effect on rBoNT/A (0)-dependent afferent inhibition. Again, TNP-ATP alone did not affect distension-induced afferent firing, a finding similar to that reported by Yu & De Groat (2008), who found that both TNP-ATP and PPADS did not influence afferent firing under basal conditions [40]. When co-applied with rBoNT/A (0), TNP-ATP partially reversed the effect, as the afferent firing was significantly lower than the control but not reduced to the extent achieved by rBoNT/A (0) alone. These findings are tentative pharmacological explorations of the potential mechanisms behind the rBoNT/A (0)-dependent effects on sensory signalling, and, while interesting, we recognise that further studies are necessary to confirm these links; however, the data collected do suggest that the sensory inhibition produced by the inactive toxin can be manipulated pharmacologically and may involve the nitrergic and purinergic signalling pathways, raising the possibility that rBoNT/A (0) could also be utilised for the treatment of conditions where nitrergic and/or purinergic signalling is disrupted, including erectile dysfunction (see [41] for a review on use of botulinum toxins as a treatment option for ED), pain and inflammation.

### 3.3. The Effect of Light Chains of rBoNT/A and rBoNT/A (0) without Cell Binding or Translocation Activities

Another important finding of this study was the inhibitory activity of the BoNT light chains, rLC/A (catalytically active LC) and rLC/A (0) (catalytically inactive LC), on the distension responses of afferent nerves. Both rLC/A and rLC/A (0) inhibited sensory firing from the bladder, but, similar to the full-length proteins, we also found that the rLC/A (0) fragment caused a greater inhibition in nerve firing than the rLC/A fragment. This suggests that the silencing of sensory signalling by BoNT/A involves a mechanism(s) that lies within the light chain of BoNT/A and is not dependent on the full-length molecule. 

When used clinically, BoNT/A preparations are normally injected into the site of interest, and the heavy chain of the toxin binds selectively and irreversibly to high-affinity receptors on the presynaptic surface. A ‘double-receptor’ mechanism has been described [42], where BoNTA initially interacts with glycolipids, such as gangliosides, concentrating the toxin on the presynaptic membrane. Once anchored close to the membrane, the toxin interacts with a second glycolipid and/or protein co-receptor, triggering receptor-mediated endocytosis. For BoNTA, this is the synaptic vesicle protein SV2 (isoforms A, B and C). The necessity of a receptor binding region for cellular entry has been well documented in the literature, with Chaddock et al. (2000) reporting that the removal of the Hc domain in an LHn/A compound led to the loss of internalisation and subsequent reductions in intracellular SNAP-25 cleavage compared to that of LHn/A conjugated to wheat germ agglutinin [43]. However, interestingly, the evidence in the present study reveals that not only is the receptor binding region not necessary, but also the translocation domain is not required to see the inhibition in sensory nerve firing. This finding opens up many potential questions regarding how the rLC/A and rLC/A (0) fragments may have interacted with cells and where they work. It is also unclear how the fragments could have traversed the epithelial barrier of the bladder, which is well recognised as the most impermeable barrier in the mammalian body. 

It is possible that distending the bladder beyond a physiological pressure (i.e., to 50 mmHg) causes disruption to the bladder wall, enabling the fragments to reach structures in the sub-epithelial space (i.e., nerve terminals, interstitial cells and blood vessels) or deeper, and that the mechanism of action could be extracellular. For example, the light chain could act as a ligand, binding to a receptor on the surface of a cell to induce a downstream change in NO or ATP release. Over the past few years, interest in novel receptors for the botulinum toxin has grown. Fibroblast growth factor receptor 3 (FGFR3) has previously been identified as another receptor for BoNT/A, with studies showing that a higher FGFR3 expression in neuronal cells can lead to increased toxin internalisation [44]. However, the available evidence to date indicates that it is the heavy chain of BoNT/A that interacts with FGFR rather than the light chain [45]. Moreover, in our current studies, the heavy chain of BoNT/A when applied alone had no significant effect on nerve firing, suggesting that it was not able to either activate a neuronal signalling mechanism or enter a cell. 

It is also possible that the LC fragments took advantage of the compensatory endocytosis process, which is initiated during bladder emptying to recover plaques expressed on the surface of the epithelial cell membranes to enter the cells and work intracellularly [46]. However, once internalised, how the LC fragments enter the cytosol or how they exert an effect is unknown.

### 3.4. Clinical Implications of the Sensory Effects of rBoNT/A (0)

The concept of a non-paralytic BoNT/A that causes the direct inhibition of sensory afferent signalling has significant potential, most importantly in tackling the lack of viable non-addictive treatments for chronic pain. Over the last 20 years, the opioid crisis has spread worldwide, and, often, people struggling with addiction were introduced to opioids following injury or surgery [47]. This illustrates the necessity for the continued development of non-addictive analgesic drugs to stem the growth of opioid dependency in patients, as it is associated with a high risk of overdose and death [48]. The current use of BoNT/A for a host of conditions usually provides long-term therapeutic benefits (i.e., 1–6 months) for patients. While this currently study was limited to a few hours for logistical reasons and, of course, used animal rather than human tissues, if the sensory inhibition identified in this study were to be long-lasting and translatable, then the use of rBoNT/A (0) to treat pain and a host of other hypersensitivity conditions would be a really exciting prospect.

With regard to the treatment of bladder disorders, rBoNT/A (0) offers the potential to provide relief from urgency without the risk of bladder retention, which is an adverse effect reported by patients following an injection of conventional BoNT/A [10]. As there is currently no BoNT/A formulation approved by the FDA for interstitial cystitis/painful bladder syndrome (IC/PBS), further development of rBoNT/A (0) may provide a route to another potential treatment for this indication.

## 4. Conclusions 

This study explored the effect of a novel recombinant BoNT/A construct that has no catalytical activity on the firing of visceral sensory nerves. These data provide evidence indicating that SNAP-25 cleavage activity may not be the driver behind the sensory effects of BoNT/A reported in the literature, as losing that ability did not preclude the rBoNT/A (0) construct from potently inhibiting afferent signalling in this assay. Following continued development and research, rBoNT/A (0) may potentially be used to treat sensory disorders and pain, providing an answer for the lack of non-addictive analgesic drugs currently available for patients. 

## 5. Materials and Methods

### 5.1. BoNTs 

AbobotulinumtoxinA (aboBoNT-A) was provided by Ipsen (Milton Park, UK). Recombinant forms of BoNT/A were produced as described below.

### 5.2. BoNT/A1 Gene Synthesis 

The BoNT/A1 protein sequence (UPI0000001386) underwent reverse translation and optimisation for expression in *E. coli* (DNA 2.0, Menlo Park, CA, USA). The DNA sequence was synthesised in two parts for safety and then combined at a silent KpnI restriction site to form the coding sequence for the full-length neurotoxin. Silent 5′ NdeI and 3′ HindIII sites were incorporated and used to insert the open reading frame into a pJ401 expression plasmid (DNA2.0) to create pJ401-BoNT/A1. The amino acid numbering of the mentioned protein sequences was based on the predicted sequence of the encoded protein, including the initiating methionine.

### 5.3. Expression and Purification of BoNT/A1 

The handling of all materials containing full-length neurotoxins occurred in microbiological safety cabinets situated within restricted-access Containment Level II laboratories. pJ401-BoNT/A1 was transformed into BLR (DE3) cells, expressed as 1L cultures and grown in an animal-component-free culture medium (AF, 12 g/L phytone peptone animal free, 24 g/L yeast extract, 10 g/L glycerol, 76 mM potassium phosphate dibasic and 14 mM potassium phosphate monobasic, 0.2% glucosamine, 30 μg/mL kanamycin) in 2.5 L shake flasks. Expression cultures were inoculated with a 1:100 dilution from an overnight starter culture and incubated at 37 °C with shaking until OD600 0.5–0.6 was reached. The temperature was then lowered to 16 °C, and after equilibrating for 1 h, IPTG (final concentration of 1 mM) was introduced to induce further growth for 20 h.

Cells were harvested via centrifugation (4300× *g*, 10 min), and cell pellets were stored at −20 °C. The cell pellets were thawed at room temperature and resuspended (3 mL/g) in lysis buffer (35 mM NaCl in 50 mM Tris pH 8.0) supplemented with 10 μL of Benzonase. Cells were lysed via ultrasonication at 4 °C (Misonix 3000 sonicator, 1 cm diameter probe, 10 × 30 s pulses, power setting 4.5, output 60–80 W). The resultant lysate was clarified via centrifugation (4300× *g*, 1 h, 4 °C), and the supernatant was retained. This supernatant was adjusted to 17.5 mM NaCl, 1.1 M (NH_4_)_2_SO_4_ and 50 mM Tris pH 8.0 by adding an equal volume of 2.2 M (NH_4_)_2_SO_4_ in 50 mM Tris pH 8.0; centrifuged (4300× *g*, 60 min, 4 °C); and loaded (4 mL/min flow rate) onto three stacked 5 mL HiTrap™ Butyl HP columns (pre-equilibrated with 1.1 M (NH_4_)_2_SO_4_ in 50 mM Tris pH 8.0). The columns were washed with 15 column volumes (CV) of 1.1 M (NH_4_)_2_SO_4_ in 50 mM Tris pH 8.0, then eluted over a 20 CV linear gradient (1.1 to 0 M (NH_4_)_2_SO_4_ in 50 mM Tris pH 8.0) and collected as 10 mL fractions.

Fractions were monitored by SDS-PAGE (NuPAGE™ 4–12% Bis-Tris gels) stained for total protein (SimplyBlue™ SafeStain, Fisher Scientific UK LTD, Loughborough, UK), and those containing the target protein were pooled and stored at 4 °C overnight. Pooled material was desalted (HiPrep 26/10 Desalting column) into 50 mM Tris pH 8.0 as a series of 11 mL batches. Each batch was collected until the conductivity surpassed 3.0 mS/cm, at which point the column was re-equilibrated until the conductivity dropped back below 3.0 mS/cm for the next batch. This process was repeated until the entire pooled fraction had been buffer-exchanged. The desalted sample was loaded at 5 mL/min onto a 5 mL itrap™ Q HP column (previously washed and charged following the manufacturer’s guidelines and pre-equilibrated with 50 mM Tris pH 8.0). The column was washed with 15 CV (50 mM Tris pH 8.0) and eluted at 5 mL/min, 15 CV linear gradient (0 to 300 mM NaCl in 50 mM Tris pH 8.0), and 2.5 mL fractions were collected. Fractions were monitored by SDS-PAGE (NuPAGE™ 4–12% Bis-Tris gels) stained for total protein (SimplyBlue™ SafeStain, Fisher Scientific UK LTD, Loughborough, UK), and those containing the target protein were pooled and stored overnight at 4 °C.

The protein concentration was determined using a Bradford assay, and the sample was concentrated to 9 mL in a Vivaspin 20 centrifugal concentrator (MwCO 5000 Da, 4300× *g*, 20 min cycles at 4 °C); the protein concentration was measured again using a Bradford assay and adjusted to 0.5 mg/mL with 125 mM NaCl in 50 mM Tris pH 8.0. Endoproteinase Lys-C was added (final Lys-C concentration 0.8 μg/mL), and the sample was incubated for 20 h at 4 °C. The sample was adjusted with an equal volume of 2 M (NH_4_)_2_SO_4_ and 50 mM Tris pH 8.0 and loaded onto two stacked 1 mL HiTrap™ Phenyl HP columns at 1.0 mL/min (pre-equilibrated with 1 M (NH_4_)_2_SO_4_ in 50 mM Tris pH 8.0).

The column was washed with 10 CV of the same buffer and then eluted over a 15 CV linear gradient (from 1 M to 0 M (NH_4_)_2_SO_4_ in 50 mM Tris pH 8.0) at 2 mL/min, and 1 mL fractions were collected. Fractions were monitored by SDS-PAGE (NuPAGE™ 4–12% Bis-Tris gels) stained for total protein (SimplyBlue™ SafeStain, Fisher Scientific, UK, LTD Loughborough UK), and those containing the target protein were pooled, concentrated and diafiltered into phosphate-buffered saline (KH2PO4 1 mM, Na_2_HPO_4_ 3 mM, NaCl 155 mM, pH 7.4) (Vivaspin 20 centrifugal concentrator MwCO 5000 Da, 4300× *g*, 20 min cycles at 4 °C); the concentration was again determined using a Bradford assay and adjusted to 0.1 mg/mL before storage in aliquots at −80 °C.

### 5.4. BoNT Constructs Tested in This Study

All of the constructs used in the study were purified using the same protocol as above. For the inactivated toxins, a mutation was made that rendered them inactive in the HEXXH region of the light chain; this involved the substitutions of glutamic acid to glutamine (E224Q) and histidine to tyrosine (H227Y).

Table 1 below outlines the BoNT constructs used in this study. In a series of pilot studies using aboBoNT-A, 100 units/mL was shown to be the dose with the most robust and reproducible response. The concentration of BoNT/A in 100 U/mL was calculated to be 3.6 pM, so this concentration was used in the experiments with other toxins to facilitate comparison ([49]). All BoNTs were dissolved in PBS and applied to the bladders using an infusion pump to fill the bladder to a pressure of 50 mmHg.

### 5.5. Animals 

These studies were performed using adult C57BL/6J mice between 8 and 12 weeks old (Jackson Laboratory, obtained from Charles River, Harlow UK). The mice were sacrificed in accordance with Schedule 1 of the Animals (Scientific Procedures) Act UK 1986. Experiments performed were in accordance with the ethical approval obtained from the UCLan Animal Welfare and Ethics Review Board (AWERB) (reference RE/16/11). 

### 5.6. Mouse Phrenic Nerve Hemidiaphragm Preparation

Hemidiaphragm recordings were performed as previously described [50]. Briefly, the left hemidiaphragm and phrenic nerve were dissected and transferred into an organ bath (Emka Technologies, Paris, France). The muscle was connected to a transducer to facilitate the recording of muscle contraction. Electrical stimulation was applied to the phrenic nerve (frequency = 1 Hz; width = 20 μs; 10 V) so as to generate a sustained force of contraction of at least 0.5 g. Following stabilisation and control steps, the sample was incubated with 100 pM recombinant BoNT/A (0) (rBoNT/A (0)), 100 pM recombinant BoNT/A1 (rBoNT/A1) or a buffer control (0.1% BSA in PBS, Sigma-Aldrich, Gillingham, UK) for three hours. Potency is expressed as the time taken to reach half paralysis (t50) using a non-linear regression analysis (GraphPad Prism v8.3). All data are expressed as mean +/− SEM; N corresponds to the number of animals.

### 5.7. Ex Vivo Mouse Bladder Electrophysiology

Nerve recordings were performed as previously described [12,51,52,53,54,55,56,57]. Briefly, the whole pelvic region of the mouse, including the lower vertebrae and urinary tract (kidneys, ureters, bladder, pelvic nerves, urethra), was dissected and placed in an organ bath continually perfused with Krebs solution (NaCl 118.4 mmol/L, NaHCO_3_ 24.9 mmol/L, CaCl_2_ 1.9 mmol/L, MgSO_4_ 1.2 mmol/L, KCl 4.7 mmol/L, KH_2_PO_4_ 1.2 mmol/L, glucose 11.7 mmol/L; obtained from Sigma-Aldrich), gassed with carbogen (95% O_2_, 5% CO_2_) and kept at ~35 °C. Ureters were tied with suture, and the urethra and the dome of the bladder were catheterised and tied with suture. The urethral catheter was attached to an infusion pump to allow the graded filling of the bladder with phosphate-buffered saline (PBS; Sigma-Aldrich) to 50 mmHg. Ramp distension to a maximal 50 mmHg pressure was chosen, as it has been previously documented to activate both low- and high-threshold afferents, which respond to both innocuous and noxious stimuli [51]. Once that pressure was reached, the tap attached to the dome catheter was opened to allow the evacuation of fluid. The pelvic and hypogastric nerve fibres emerging from the bladder were dissected, cut and inserted into a suction glass electrode connected to a Neurolog headstage and AC amplifier. The signals were amplified and filtered and sent to a spike processor, which discerned signals from noise through a threshold set by the researcher. The spike processor counted the nerve impulses induced by bladder distension. 

### 5.8. Experimental Protocols

The distension protocol followed in the bladder preparations is shown below in Figure 10. At the beginning of each experiment, the bladder was distended three times using PBS to ensure reproducibility, after which the infusion syringe was changed to one containing BoNT solution, and three more distensions were performed. After this, the syringe was replaced with one containing PBS, and distensions continued for 90 min to assess the effect of BoNT on pelvic and hypogastric sensory nerve firing. Any BoNT within the intraluminal fluid was deactivated using Presept (Advanced Sterilization Products) as it came out of the dome catheter.

### 5.9. Drugs

N(G)-Nitro-L-arginine methyl ester (L-NAME) and 2′,3′-O-trinitrophenyl ATP (TNP-ATP) were obtained from Tocris (Oxford, UK), diluted in distilled water and stored at −20 °C.

### 5.10. Data Analysis

The afferent nerve activity recorded from multi-fibre nerve bundles was captured and counted using a Digitimer D130 spike processor, which detected the number of field potentials that crossed a pre-set threshold. Distension responses were analysed using a Spike2 script, and nerve activity at baseline was subtracted from peak firing at 50 mmHg. To analyse the effect of BoNT on distension-induced firing, responses at 30, 60 and 90 minutes post-BoNT treatment were normalised to the control distension before the addition of BoNT. Afferent nerve data are expressed as normalised afferent firing in impulses per second (imps/sec), and bladder compliance was assessed from the pressure–volume relationship during the distension of the bladder. All data are expressed as mean +/− SEM and n number, which refers to the number of animals. A statistical analysis was conducted using GraphPad Prism v.8.0.1, using Student’s *t* tests and one-way and two-way ANOVA where appropriate, with a Bonferroni post hoc test. A *p* value <0.05 was designated to be statistically significant.

### 5.11. Immunohistochemistry

At the end of the nerve recording, the bladders treated with the vehicle and abobotulinumtoxinA (aboBoNT-A) were fixed at room temperature (~20 °C) in 4% paraformaldehyde (VWR Chemicals, Luton UK) and then kept in 1% PFA/PBS solution at 4 °C. Four aboBoNT-A-treated and four PBS-treated bladders were included in this analysis, dissected from separate mice. The bladder samples were sent to be processed, blocked and sectioned to 8 µm at the University College London pathology lab. Once received, the slides were blinded with tape and numbered so that corresponding sections could be compared between bladders. 

Immunohistochemical staining was performed using an avidin–biotin–peroxidase protocol as previously described [18,58]. Sections to be incubated with the anti-SNAP-25 antibody did not receive antigen retrieval, whereas those to be stained with the anti-cleaved SNAP-25 (anti-cSNAP-25) antibody were immersed in EDTA buffer (pH 9, Sigma-Aldrich, Gillingham, UK, E9884) at 98 °C. All sections were first blocked with 3% hydrogen peroxide (Sigma-Aldrich, Gillingham, UK, 1.07209), followed by 10% horse serum (Sigma-Aldrich, H1270) in Dako buffer. The sections were incubated at room temperature for one hour with anti-SNAP-25 (rabbit, 1/1000, Sigma-Aldrich, Gillingham, UK, S9684) and overnight with anti-cSNAP-25 (rabbit, 1/1000, provided by the research Pathology & Safety Biomarker Department, Ipsen Bioinnovation, Les Ulis, France). All sections (both primary antibodies) were incubated with a secondary biotinylated antibody (horse, 1/400, Vector Labs, Peterbrough, UK, BA-1100) for 30 min at room temperature. To amplify signals, Vectastain Elite ABC HRP Reagent (Vector Labs, PK-7100) was applied for 30 min at room temperature, followed by haematoxylin (Sigma-Aldrich, Gillingham, UK, H2126). Finally, slides were dehydrated with ethanol and xylene substitute Histochoice Clearing Agent (Sigma-Aldrich, Gillingham, UK, H2779). All slides were imaged using a light microscope, and the resulting image files were analysed in quadrants but not quantified, as preliminary analyses showed no cSNAP-25 staining in any of the bladders tested. 

## 6. Patents

WO 2022/208091.

## Figures and Tables

**Figure 1 toxins-16-00030-f001:**
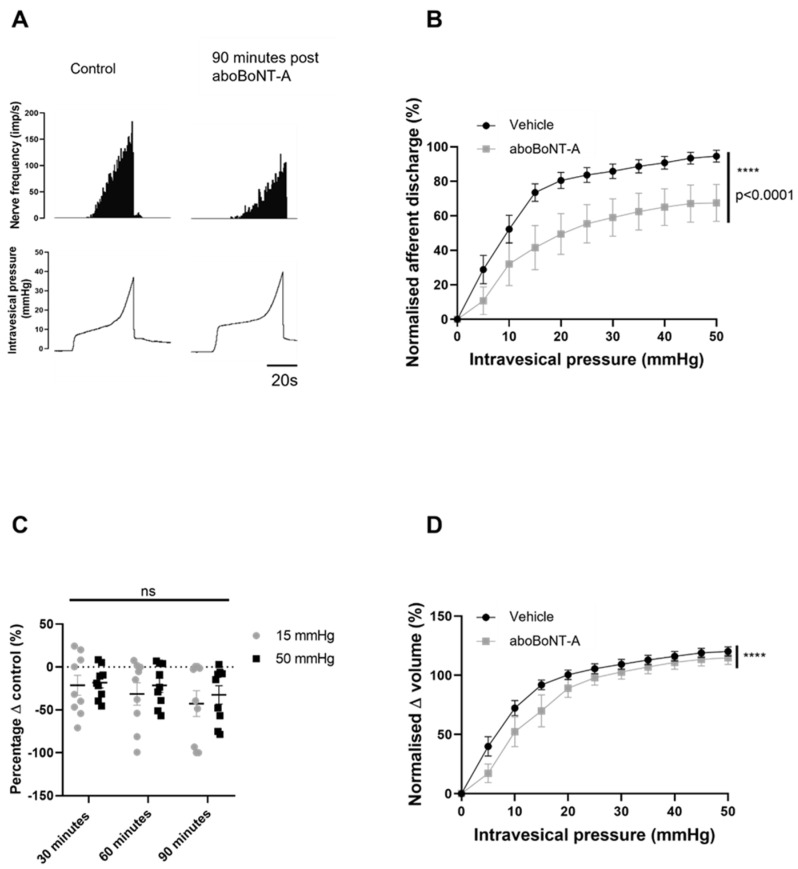
aboBoNT-A directly inhibited mechanosensitive firing. (**A**) A representative trace showing the response of afferent nerves to distension prior to (control) and 90 min after application of 100 U/mL aboBoNT-A. (**B**) Afferent responses to distension were significantly reduced in preparations that received intravesical aboBoNT-A (n = 9) compared to vehicle-treated (n = 13) preparations (*p* < 0.0001; two-way ANOVA). (**C**) Percentage change in peak firing at 15 and 50 mmHg showed no difference between the early and later time points (*p* = 0.1003; n = 9; two-way ANOVA). (**D**) The pressure–volume relationship was significantly lower in preparations that received intravesical aboBoNT-A (n = 9) than in vehicle-treated (n = 13) preparations (**** = *p* < 0.0001; two-way ANOVA).

**Figure 2 toxins-16-00030-f002:**
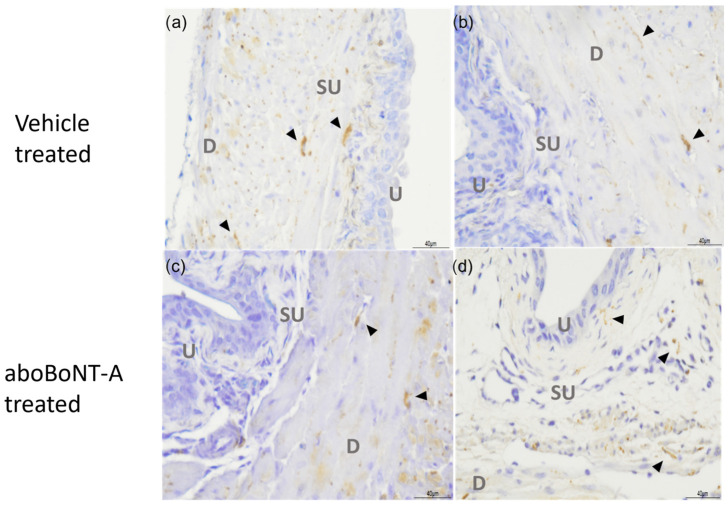
SNAP-25 immunoreactivity in the bladder wall. A representative figure showing vehicle-treated (**a**,**b**) and aboBoNT-A-treated (**c**,**d**) bladder samples labelled with anti-SNAP-25 antibody. SNAP-25 immunoreactivity is labelled by black arrowheads, found predominantly in the suburothelial and detrusor layers. Images labelled as U—urothelium, SU—suburothelium and D—detrusor smooth muscle.

**Figure 3 toxins-16-00030-f003:**
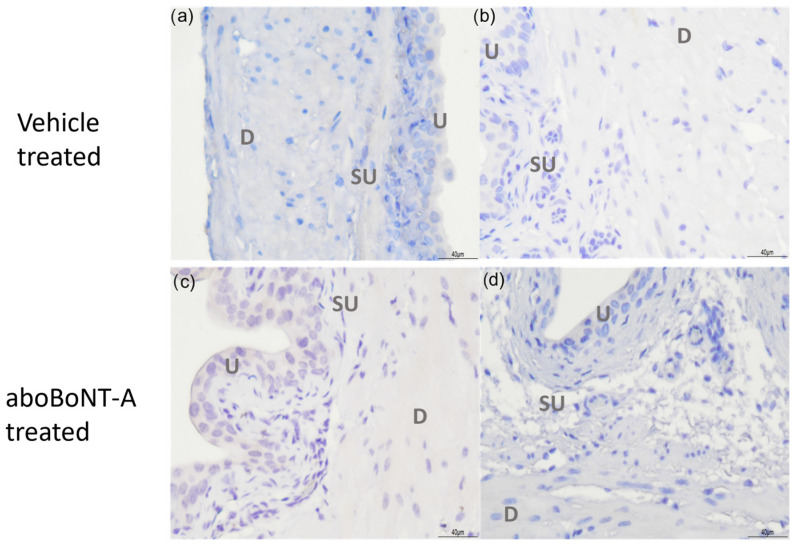
cSNAP-25 immunoreactivity within the bladder wall. A representative figure showing vehicle-treated (**a**,**b**) and aboBoNT-A-treated (**c**,**d**) bladder samples labelled with anti-cSNAP-25 antibody. No positive staining was found in vehicle- or aboBoNT-A-treated samples. Images labelled as U—urothelium, SU—suburothelium and D—detrusor smooth muscle.

**Figure 4 toxins-16-00030-f004:**
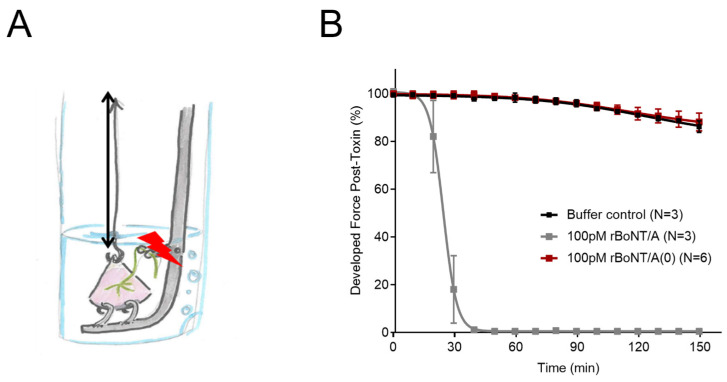
The mouse phrenic nerve hemidiaphragm assay characterising the effects of rBoNT/A and rBoNT/A (0). (**A**) A schematic diagram showing the experimental setup of the preparation. (**B**) The effects of rBoNT/A (n = 3) and rBoNT/A (0) (n = 6) on hemidiaphragm contractility. rBoNT/A1 significantly and completely blocked the developed force after 20 min, which did not recover (*p* < 0.0001 2-way ANOVA).

**Figure 5 toxins-16-00030-f005:**
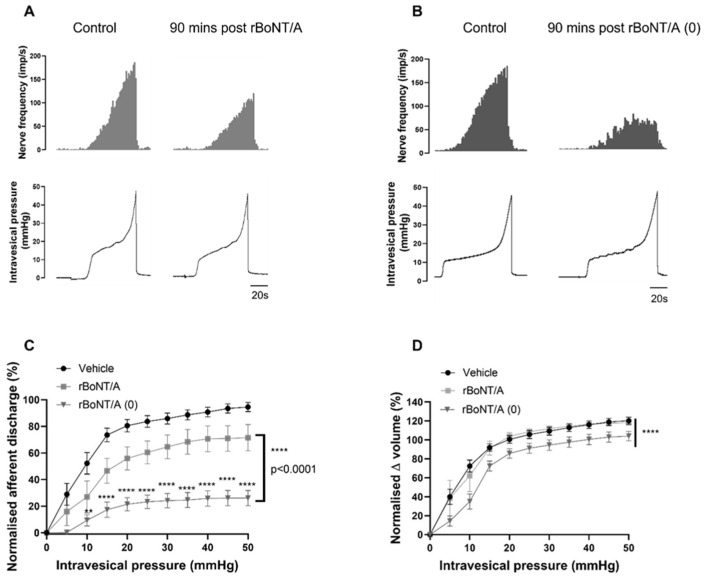
The effects of recombinant active rBoNT/A and inactive rBoNT/A (0) on distension-induced afferent signalling. (**A**) A representative trace showing the response of afferent nerves to distension prior to (control) and 90 min after rBoNT/A treatment. (**B**) A representative trace showing the response of afferent nerves to distension prior to (control) and 90 min after rBoNT/A (0) treatment. (**C**) Distension-induced afferent responses were significantly reduced by rBoNT/A (**** *p* < 0.0001; n = 6; two-way ANOVA) and rBoNT/A (0) (**** *p* < 0.0001; n = 8; two-way ANOVA) compared to vehicle-treated preparations (n = 13). The distension-induced firing in preparations treated with rBoNT/A (0) was significantly lower than in those treated with rBoNT/A (**** *p* < 0.0001) (** *p* < 0.01). (**D**) The pressure–volume relationship of rBoNT/A-treated preparations was not different to that of vehicle-treated preparations (*p* = 0.8765; n = 6; two-way ANOVA), while rBoNT/A (0) significantly reduced the pressure–volume relationship (**** *p* < 0.0001; n = 8; two-way ANOVA).

**Figure 6 toxins-16-00030-f006:**
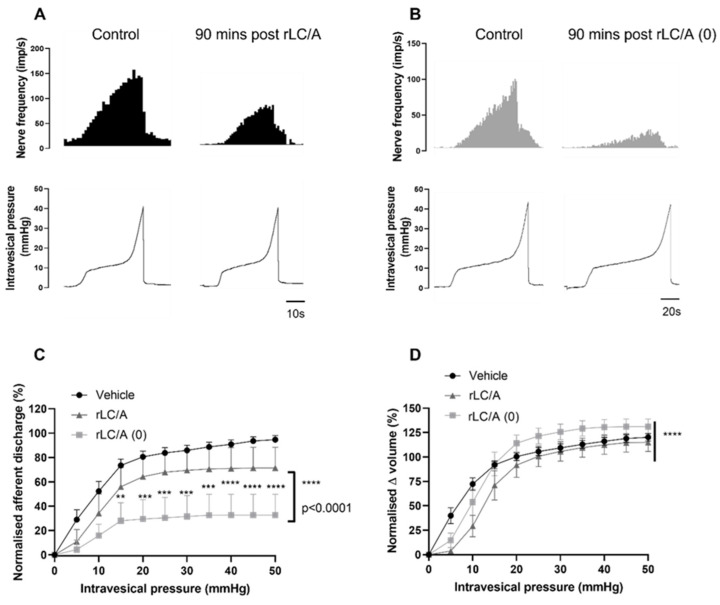
The effect of active (LC/A) and inactive (LC/A (0)) light chains on distension-induced afferent signalling. (**A**) A representative trace showing the response of afferent nerves to distension prior to (control) and 90 min after rLC/A treatment. (**B**) A representative trace showing the response of afferent nerves to distension prior to (control) and 90 min after rLC/A (0) treatment. (**C**) Distension-induced afferent responses were significantly reduced by rLC/A (**** *p* < 0.0001; n = 5; two-way ANOVA) and rLC/A (0) (**** *p* < 0.0001; n = 5; two-way ANOVA). The distension-induced firing in preparations treated with rLC/A (0) was significantly lower than in those treated with rLC/A (**** *p* < 0.0001 two-way ANOVA). (**D**) The pressure–volume relationship of rLC/A-treated preparations was significantly lower than that of vehicle-treated preparations (**** *p* < 0.0001; n = 5; two-way ANOVA), while rLC/A (0) was not significantly different (*p* = 0.065; n = 5; two-way ANOVA) (** *p* < 0.01; *** *p* < 0.001).

**Figure 7 toxins-16-00030-f007:**
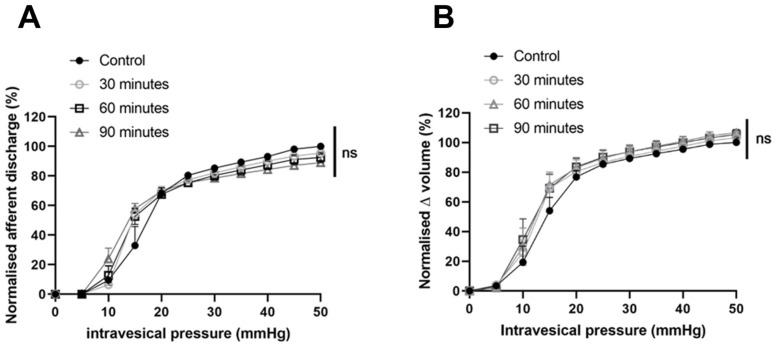
HC/A had no effect on distension-induced firing. (**A**) Afferent responses to distension were unchanged 30, 60 and 90 min after treatment (*p* = 0.9819, n = 4, two-way ANOVA. (**B**) There was no difference in the pressure–volume relationship after intravesical HC/A compared to control (*p* = 0.0513; n = 4; two-way ANOVA).

**Figure 8 toxins-16-00030-f008:**
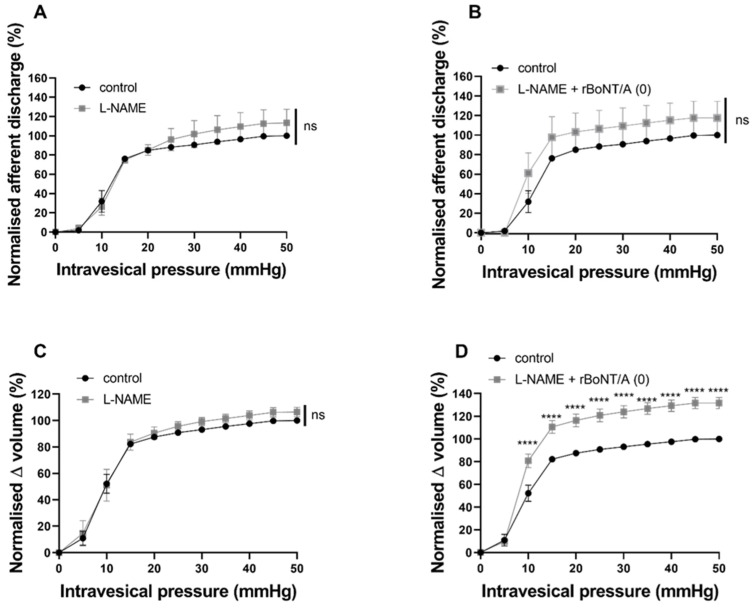
The effect of L-NAME pre-treatment on the inhibitory effect of rBoNT/A (0). (**A**) L-NAME treatment alone did not alter distension-induced afferent firing, as the response to intravesical L-NAME was not different to that of control (*p* = 0.475; n = 5; two-way ANOVA). (**B**) Afferent responses to distension of the bladder following treatment with L-NAME alongside rBoNT/A (0) were not different to the afferent response to controlled distension of the bladder (*p* = 0.3881; n = 5; two-way ANOVA). (**C**) The pressure–volume relationship was not altered by intravesical L-NAME alone (*p* = 0.475; n = 5; two-way ANOVA). (**D**) The pressure–volume relationship was significantly increased following treatment with L-NAME alongside rBoNT/A (0) compared to control responses (**** *p* < 0.0001; n = 5; two-way ANOVA). Sidak’s multiple comparisons test revealed significant increases in volume from 10 mmHg (**** *p* < 0.0001) to 50 mmHg (**** *p* < 0.0001).

**Figure 9 toxins-16-00030-f009:**
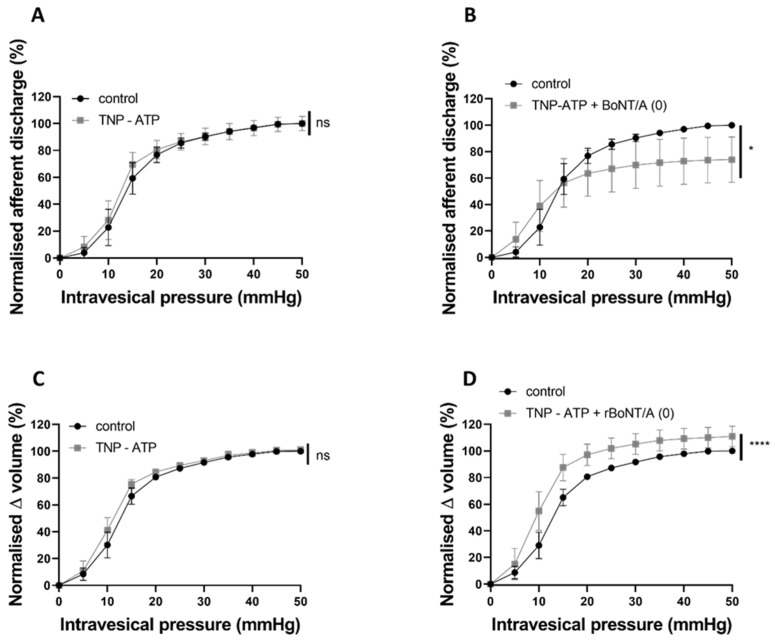
The effect of TNP-ATP pretreatment on the inhibitory effect of rBoNT/A (0). (**A**) The afferent response to intravesical TNP-ATP was not different to that of control (*p* = 0.4396; n = 6; two-way ANOVA). (**B**) The afferent response to distension following TNP-ATP applied alongside rBoNT/A (0) was significantly lower than that of control (* *p* = 0.0301; n = 6; two-way ANOVA). (**C**) The pressure–volume relationship was not altered by intravesical TNP-ATP alone (*p* = 0.2868; n = 6; two-way ANOVA). (**D**) The pressure–volume relationship was significantly increased by TNP-ATP applied alongside rBoNT/A (0) (**** *p* < 0.0001; n = 6; two-way ANOVA).

**Figure 10 toxins-16-00030-f010:**
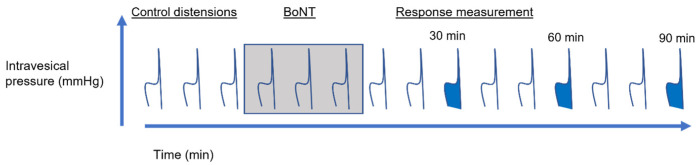
A schematic diagram of the distension protocol. The bladder was distended to 50 mmHg every 10 min throughout the protocol. ‘Control’ distensions were performed using intravesical PBS; BoNT was applied for three distensions (in PBS, shaded), after which time bladders were then distended again with PBS. The filled distensions in blue correspond to the time points analysed.

**Table 1 toxins-16-00030-t001:** A diagram describing the composition of all of the BoNT/A constructs used in this study.

Name	Structure	Description	Method of Production
aboBoNT-A	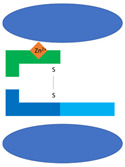	Whole BoNT/A complex, including the non-toxic accessory proteins (NAPs)	Purified from C. botulinum
rBoNT/A	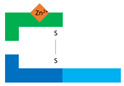	Full-length BoNT/A	Recombinant; expressed in *E. coli* and purified
rLC/A	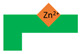	Light chain only	Recombinant; expressed in *E. coli* and purified
rHC/A	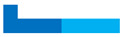	Heavy chain only	Recombinant; expressed in *E. coli* and purified
rBoNT/A (0)	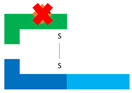	Full-length mutated BoNT/A, catalytically inactive	Recombinant; expressed in *E. coli* and purified
rLC/A (0)	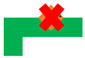	Mutated light chain only, catalytically inactive	Recombinant; expressed in *E. coli* and purified

## Data Availability

Datasets can be made available on request from the authors.

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
