# Peer review of "A Novel Catalytically Inactive Construct of Botulinum Neurotoxin A (BoNT/A) Directly Inhibits Visceral Sensory Signalling"

_toxins, 2024, doi:10.3390/toxins16010030_

Round 1
Reviewer 1 Report
Comments and Suggestions for Authors
Botulinum neurotoxin A (BoNT/A), a potent neurotoxin disrupting cholinergic neurotransmission via SNAP-25 cleavage, is known for its paralytic effects and potential to inhibit sensory nerve activity. However, the underlying mechanisms remain unclear. This study aimed to decipher how BoNT/A affects afferent signaling from the bladder.
To explore the role of SNAP-25 cleavage in BoNT/A-induced sensory signal inhibition, they engineered a non-paralyzing form of BoNT/A with an inactive light chain, termed rBoNT/A (0). Additionally, they created recombinant light chain (LC) domain proteins to examine entry mechanisms, given that the heavy chain (HC) of BoNT/A is responsible for the light chain's internalization.
Introduction
please correct following
1) "The BoNT/A protein is composed of a heavy chain (HC) of 100 kDa, and light chain (LC) a 50 kDa, with functionally distinct properties." To correct the sentence and make it grammatically accurate, it should be revised as follows: "The BoNT/A protein is composed of a heavy chain (HC) of 100 kDa and a light chain (LC) of 50 kDa, each with functionally distinct properties."
Material and Method
2) Change “The BoNT/A1 protein sequence (UPI0000001386) was back-translated and codon-optimized for expression in E. coli (DNA 2.0, Menlo Park, CA, USA).” To “The BoNT/A1 protein sequence (UPI0000001386) underwent reverse translation and optimization for expression in E. coli (DNA 2.0, Menlo Park, CA, USA).”
3) Change “The DNA sequence was synthesised in two parts, for safety, and subsequently combined, at a silent KpnI restriction site, to create a coding sequence for the full-length neurotoxin.” To “The DNA sequence was synthesized in two parts for safety and then combined at a silent KpnI restriction site to form the coding sequence for the full-length neurotoxin.”
4) Change “Amino acid numbers of the protein sequences cited are counted from the predicted sequence of the encoded protein, including the initiating methionine.” And “The amino acid numbering of the mentioned protein sequences is based on the predicted sequence of the encoded protein, including the initiating methionine.”
5) Change “All manipulations of material containing full-length neurotoxin were performed in microbiological safety cabinets, located in restricted-access Containment Level II laboratories.” To “All handling of materials containing full-length neurotoxin occurred in microbiological safety cabinets situated within restricted-access Containment Level II laboratories.”
6) Change “The temperature was reduced to 16 °C and the cultures equilibrated for 1 h, then induced with IPTG (final concentration of 1 mM) and grown for a further 20 h.”
7) “The temperature was then lowered to 16 °C, and after equilibrating for 1 hour, IPTG (final concentration of 1 mM) was introduced to induce further growth for 20 hours.”
Discussion
8) Consider changes in “The study showed that a catalytically inactive BoNT/A construct caused profound inhibition of sensory nerve firing from the bladder. The inhibitory effect was significantly greater than that produced by a catalytically active recombinant BoNT/A toxin or the full BoNT/A complex.” To “The research demonstrated that a catalytically inactive BoNT/A construct led to a profound inhibition of sensory nerve firing from the bladder, which was notably more significant than the effect produced by a catalytically active recombinant BoNT/A toxin or the complete BoNT/A complex. This unexpected finding has substantial implications for the clinical efficacy and potency of BoNT/A, particularly in conditions where sensory inhibition (e.g., pain and hypersensitivity) is beneficial.”
9) Also, “The study found that both recombinant light chain (rLC/A) and catalytically inactive recombinant light chain (rLC/A (0)) significantly reduced afferent signaling without the presence of the heavy chain responsible for cell binding and entry.” To “Both the recombinant light chain (rLC/A) and the catalytically inactive recombinant light chain (rLC/A (0)) exhibited the capability to significantly reduce afferent signaling, despite the absence of the typical heavy chain responsible for cell binding and entry. This suggests that the inhibition of sensory nerves by BoNT/A may not be solely dependent on SNARE cleavage but may involve other activities within the light chain region.”
10) Additionally, “The study revealed that intravesical instillation of aboBoNT-A into the mouse bladder led to a significant reduction in distension-induced afferent firing, aligning with findings reported by Collins et al. (2013).” To “The research showed that intravesical instillation of aboBoNT-A into the mouse bladder resulted in a notable decrease in distension-induced afferent firing, consistent with previous findings reported by Collins et al. (2013). However, the precise mechanisms underlying BoNT/A-mediated modulation of bladder sensory signaling remain elusive”
Overall, their findings revealed that despite lacking catalytic activity, rBoNT/A (0) remarkably suppressed afferent responses to bladder distension to a greater extent than the catalytically active rBoNT/A. Similarly, the inactive rLC/A (0) significantly inhibited afferent responses more than the active rLC/A protein. Immunohistochemistry results showed negative cleaved SNAP-25 staining. Moreover, the use of purinergic and nitrergic antagonists partly and completely reversed sensory inhibition, respectively.
I would recommend the authors change the minor errors.
Comments on the Quality of English LanguageNothing to comment about.
Author Response
Reviewer 1:
Dear reviewer 1: The authors would like to thank-you for your rigorous and detailed review of our manuscript and for the suggested revisions to the text, we have implicated all of your changes, and this has only served to improve the quality of the writing. We have addressed each of your suggestions in a point-by-point response below, best wishes Dr Daly.
Introduction
- "The BoNT/A protein is composed of a heavy chain (HC) of 100 kDa, and light chain (LC) a 50 kDa, with functionally distinct properties." To correct the sentence and make it grammatically accurate, it should be revised as follows: "The BoNT/A protein is composed of a heavy chain (HC) of 100 kDa and a light chain (LC) of 50 kDa, each with functionally distinct properties."
Thank you for the suggested revision, this has been amended and is highlighted in blue
Material and Method
2) Change “The BoNT/A1 protein sequence (UPI0000001386) was back-translated and codon-optimized for expression in E. coli (DNA 2.0, Menlo Park, CA, USA).” To “The BoNT/A1 protein sequence (UPI0000001386) underwent reverse translation and optimization for expression in E. coli (DNA 2.0, Menlo Park, CA, USA).”
Thank you for the suggested revision, this has been amended and is highlighted in blue
3) Change “The DNA sequence was synthesised in two parts, for safety, and subsequently combined, at a silent KpnI restriction site, to create a coding sequence for the full-length neurotoxin.” To “The DNA sequence was synthesized in two parts for safety and then combined at a silent KpnI restriction site to form the coding sequence for the full-length neurotoxin.”
Thank you for the suggested revision, this has been amended and is highlighted in blue
4) Change “Amino acid numbers of the protein sequences cited are counted from the predicted sequence of the encoded protein, including the initiating methionine.” To “The amino acid numbering of the mentioned protein sequences is based on the predicted sequence of the encoded protein, including the initiating methionine.”
Thank you for the suggested revision, this has been amended and is highlighted in blue
5) Change “All manipulations of material containing full-length neurotoxin were performed in microbiological safety cabinets, located in restricted-access Containment Level II laboratories.” To “All handling of materials containing full-length neurotoxin occurred in microbiological safety cabinets situated within restricted-access Containment Level II laboratories.”
Thank you for the suggested revision, this has been amended and is highlighted in blue
6) Change “The temperature was reduced to 16 °C and the cultures equilibrated for 1 h, then induced with IPTG (final concentration of 1 mM) and grown for a further 20 h.” to “The temperature was then lowered to 16 °C, and after equilibrating for 1 hour, IPTG (final concentration of 1 mM) was introduced to induce further growth for 20 hours.”
Thank you for the suggested revision, this has been amended and is highlighted in blue
Discussion
8) Consider changes in “The study showed that a catalytically inactive BoNT/A construct caused profound inhibition of sensory nerve firing from the bladder. The inhibitory effect was significantly greater than that produced by a catalytically active recombinant BoNT/A toxin or the full BoNT/A complex.” To “The research demonstrated that a catalytically inactive BoNT/A construct led to a profound inhibition of sensory nerve firing from the bladder, which was notably more significant than the effect produced by a catalytically active recombinant BoNT/A toxin or the complete BoNT/A complex. This unexpected finding has substantial implications for the clinical efficacy and potency of BoNT/A, particularly in conditions where sensory inhibition (e.g., pain and hypersensitivity) is beneficial.”
Thank you for the suggested revision, in this paragraph we are trying to say that the inhibition was not just statistically significant but that the magnitude of the effect was greater, a therefore I am concerned about using the term “notably more significant’ as this could be misleading, however we have amended the paragraph in light of your suggestions, the amendments are highlighted in blue (lines 338-343)
9) Also, “The study found that both recombinant light chain (rLC/A) and catalytically inactive recombinant light chain (rLC/A (0)) significantly reduced afferent signaling without the presence of the heavy chain responsible for cell binding and entry.” To “Both the recombinant light chain (rLC/A) and the catalytically inactive recombinant light chain (rLC/A (0)) exhibited the capability to significantly reduce afferent signaling, despite the absence of the typical heavy chain responsible for cell binding and entry. This suggests that the inhibition of sensory nerves by BoNT/A may not be solely dependent on SNARE cleavage but may involve other activities within the light chain region.”
Thank you for the suggested revision, this has been amended and is highlighted in blue
10) Additionally, “The study revealed that intravesical instillation of aboBoNT-A into the mouse bladder led to a significant reduction in distension-induced afferent firing, aligning with findings reported by Collins et al. (2013).” To “The research showed that intravesical instillation of aboBoNT-A into the mouse bladder resulted in a notable decrease in distension-induced afferent firing, consistent with previous findings reported by Collins et al. (2013). However, the precise mechanisms underlying BoNT/A-mediated modulation of bladder sensory signaling remain elusive”
Thank you for the suggested revision, this has been amended and is highlighted in blue
Reviewer 2 Report
Comments and Suggestions for Authors
Dear authors,
The work is well presented and interesting, however some clarifications are required to support the conclusions.
Abstract:
--------------------
Line 7: please modify "poorly understood" since many studies have already investigated the mecanisms underlying the effects of BoNT/A on pain.
Line 21: remove "the" after "classical"
The absence of cleavage needs further confirmation as imaging is not enough to prove absence of cleaved form of SNAP-25.
Introduction:
-----------------
Line 30: asphyxiation is not adequate since this is caused by external agents, it is respiratory distress subsequent to diaphragm paralysis.
Lines 61-67: please quote existing published work already exploring the mecanisms behind the sensory effects of BoNT/A.
Lines 69-70: the sentence is too affirmative since there is a need to confirm the true absence of cleaved SNAP-25
Results:
---------
Lines 87-88: please justify how this effect on firing may not be the indirect consequence of the action of BoNT/A on the bladder wall.
Lines 114-116: the immunochemistry is not sufficient to prove absence of cleaved SNAP-25. need to add further confirmation using western blot or other molecular explorations.
Line 150: "caused by"
Line 282: please briefly explain in this paragraph why the NOS inhibitor may interfere with BoNT/A action and propose a mecanism in the discussion.
Line 290: please propose a mecanism underlying the role of NO in BoNT/A action.
Line 310-314: idem: please propose the mecanism underlying the interaction with TNP-ATP.
Line 340: figure 9D: alongside rBoNT/A (0)
Discussion:
-------------
Line 353: please add that this could also be due to action on other SNAP proteins like SNAP-23. The absence of cleaved SNAP-25 does not prove absence of cleavage of other SNAPs.
Line 370-377: the absence of cleavage needs further confirmation since the immunochemistry is not precise enough. Also, please note that the cleavage level is not directly related to alteration of the function and cannot be considered as the only parameter explaining BoNT/A actions.
Line 428: absence of SNAP-23 cleavage needs confirmation and cannot be assumed as absence of SNAP-25 cleavage also needs confirmation.
Line 469: studies on erectile dysfunction already exist and need to be quoted.
Author Response
Dear reviewer 2: The authors would like to thank you for your rigorous and detailed review of our manuscript and for the suggested revisions to the text, we have amended the manuscript according to your suggestions and we have provided answers to the questions posed in your review. We hope that our responses and rationale are satisfactory. We have addressed each of your comments in a point-by-point response below, best wishes Dr Daly
Abstract:
Line 7: please modify "poorly understood" since many studies have already investigated the mechanisms underlying the effects of BoNT/A on pain.
Thank you for the suggested revision, this has been amended and is highlighted in yellow
Line 21: remove "the" after "classical"
Thank you for the suggested revision, this has been amended and is highlighted in yellow
The absence of cleavage needs further confirmation as imaging is not enough to prove absence of cleaved form of SNAP-25.
Thank you for this comment, we take this on board and indeed agree that imaging alone is not enough to prove the absence of SNARE cleavage however in the study we also provide functional evidence to suggest that SNAP25 cleavage is indeed absent. Figure 4 shows the effect of the catalytically inactive rBoNT/A (0) on a mouse phrenic nerve hemidiaphragm assay, here we can see that the rBoNT/A (0) had no effect on hemidiaphragm contractility supporting the idea that the toxin prevents cholinergic transmission. Since its well established that ACh transmission in this assay is SNAP25 dependent it seems reasonable to assume that the mutation has prevented the toxin from cleaving SNAP-25 in its normal way.
Moreover, the rBoNT/A (0) molecule was generated by introducing mutations which disrupt coordination of the catalytic Zn2+ ion and abolishes the presumed catalytic mechanism [Breidenbach & Brunger, 2004, Nature] so it is reasonable, given that we found no cleaved SNAP-25 and that the molecule didn’t paralyse the diaphragm to assume that inactivation by this mutation was successful. Moreover, the catalytic endopeptidase activity of purified recombinant botulinum light chain A, containing these two mutations (E224Q, H227Y), has been tested directly, in a cell free assay. At the highest concentration tested (1.8 mg/ml), there was no measurable endopeptidase activity, compared to the native light chain sequence (which was a theoretical minimum of 10^7 fold more active in the assay) [Shone and others, 2009, Infect Immun]. We have added a citation of this reference (Results line 145), highlighted in yellow
Introduction:
Line 30: asphyxiation is not adequate since this is caused by external agents, it is respiratory distress subsequent to diaphragm paralysis.
Thank you for pointing this out I have amended the sentence to remove any confusion (Line 632-633), changes are highlighted in yellow
Lines 69-70: the sentence is too affirmative since there is a need to confirm the true absence of cleaved SNAP-25
Thank you for this comment, in this sentence we chose to use the phrase “ without evidence of cleaved SNAP-25” (instead of something more affirmative such as “in the absence of SNAP25 cleavage”), we believe this sentence is an accurate representation of what this study found.
As already addressed in an earlier point, we investigated SNAP-25 cleavage using both imaging and functional experiments and while we cannot definitively state that SNAP-25 cleavage was absent when using the rBoNT/A (0), we did not find any evidence for it. In line 379-381 the limitations of the imaging studies are also addressed.
Results:
Lines 87-88: please justify how this effect on firing may not be the indirect consequence of the action of BoNT/A on the bladder wall.
This is an excellent point and one we have also considered, however the effect on bladder compliance following application of aboBoNT-A is very small yet this is concomitant with a relatively large effect on sensory firing (25-30%) which suggests that the two phenomenon are uncoupled, especially given that in the higher pressures there were minimal changes in compliance but maximal changes in nerve firing.
Moreover, mechanosensitive afferents innervating hollow muscular organs such as the bladder typically response to distortion and tone of the bladder wall (i.e. they are classically considered to be tension receptors and all of our previous work on afferent function would confirm this assumption). In these studies application of aboBoNT-A caused the pressure-volume relationship to go down, as such a smaller volume of fluid was needed to generate a given change in pressure (i.e. to generate 10mmHg pressure the bladder was filled to 75% of its total volume before the toxin but 50% after the toxin), this suggest that the bladder is less complaint and has a higher tone after the toxin is applied. Since the afferents respond to tension we would therefore expect afferent discharge to increase at this pressure and not fall, but in fact we see the opposite. Why we saw these small changes in bladder tone is not clear from this study and is something that warrants further investigation.
Lines 114-116: the immunochemistry is not sufficient to prove absence of cleaved SNAP-25. need to add further confirmation using western blot or other molecular explorations.
Thank you for this comment, as addressed by an earlier point we also had some functional data to support the immunohistochemistry data- so our conclusions are not based on the immunohistochemical data alone. However, we do take on board the point that neither are conclusive. Since western blot analysis would use a similar approach (i.e. antibody detection) we are not convinced this would present an advantage over the immunohistochemical analysis which would also provide insight into where expression occurs. However, we will consider this carefully for any further studies with the rBoNT (0). We have also made changes in the discussion section of the manuscript to highlight the possibility that rBoNT/A (0) may retain some level of proteolytic activity that we did not detect. Changes highlighted in yellow
Line 150: "caused by"
Thank you for the suggested revision, this has been amended and is highlighted in yellow
For clarity we will address the following points together:
- Line 282: please briefly explain in this paragraph why the NOS inhibitor may interfere with BoNT/A action and propose a mechanism in the discussion.
- Line 290: please propose a mechanism underlying the role of NO in BoNT/A action.
- AND Line 310-314: idem: please propose the mecanism underlying the interaction with TNP-ATP
Thank you for these comments, unfortunately we do not have a mechanism of action to propose at this point. The data reported in this study suggests that due to the loss of catalytic activity, rBoNT/A (0) may act through an alternative mechanism of action, different from the classical mechanism involving SNAP-25 cleavage.
The rationale behind investigating nitrergic and purinergic pathways was that following BoNT/A treatment, modulation of NO and ATP release has been shown in the literature (discussed in lines 429-431). The cleavage of SNAP-25 may not fully explain these effects as the release of these sensory mediators is partially (ATP) and wholly (NO) independent of exocytosis. Using a molecule that was unable to cleave SNAP-25, we aimed to understand if BoNT/A may still be able to modulate these pathways, and how this may translate to afferent responses to distension.
As the results show purinergic and nitrergic antagonists partially and fully restored afferent firing, we tentatively suggest that these pathways may be involved in rBoNT/A dependent inhibition. Unfortunately, we currently do not know how exactly this toxin induces the afferent inhibition, where in the bladder wall it acts, which cell type it targets or what its duration of action is. Possibly, rBoNT/A may activate nitrergic or purinergic signalling directly, or act as a ligand on a receptor and therefore have secondary effects on purinergic or nitrergic signalling mechanisms.
We certainly recognise that this is a limitation of this study and believe that it would be irresponsible to speculate on a mechanism for which we have no supporting data. We have included this work in the manuscript to show that what is happening can be pharmacologically manipulated. Future studies are now needed to determine how this catalytically inactive compound can alter sensory transmission from the bladder.
Line 340: figure 9D: alongside rBoNT/A (0)
Thank you for the suggested revision, this has been amended and is highlighted in yellow
Discussion:
Line 353: please add that this could also be due to action on other SNAP proteins like SNAP-23. The absence of cleaved SNAP-25 does not prove absence of cleavage of other SNAPs.
We have changed the phrase “SNARE cleavage” to “SNAP25 cleavage”, to make clear that our data do not exclude cleavage of other SNARE proteins such as SNAP23. We also discuss potential cleavage of SNAP23 in lines 418-427
Line 370-377: the absence of cleavage needs further confirmation since the immunochemistry is not precise enough. Also, please note that the cleavage level is not directly related to alteration of the function and cannot be considered as the only parameter explaining BoNT/A actions.
We have added the following sentence to line 381: Further studies are required to irrefutably confirm that rBoNT/a (0) does not cleave SNAP-25 using other more sensitive methodology. This is highlighted in yellow.
Line 428: absence of SNAP-23 cleavage needs confirmation and cannot be assumed as absence of SNAP-25 cleavage also needs confirmation.
The potential cleavage of SNAP23 is covered on lines 411-420
Line 469: studies on erectile dysfunction already exist and need to be quoted.
We have now made reference to Argiolas 2023 in the text. Changes highlighted in yellow
Reviewer 3 Report
Comments and Suggestions for Authors
I carefully read the manuscript; it is well-written and fascinating. The experimental design is appropriate, and the results are clearly described and commented on. In the section 'Conclusions', the authors should consider the opportunity to emphasise the spin-off of their findings on treating pathologies related to sensory disorders and pain.
Author Response
Dear reviewer 3: The authors would like to thank you for your review of our manuscript and for your positive feedback. As you have suggested, we have alluded to the wider potential impact of this work in the treatment of pathologies related to sensory disorders and pain (lines 540-547) but we have not been careful not to over emphasise this point given that this work was conducted using isolated ex vivo tissue samples from mouse and lacks in vivo data or work on any human tissue samples, making it difficult to translate the findings into any clinical setting at this point.